# EasyInv: Toward Fast and Better DDIM Inversion

Ziyue Zhang [*1]   Mingbao Lin [*2]   Shuicheng Yan [32]   Rongrong Ji [1]

## Abstract

This paper introduces EasyInv, an easy yet novel approach that significantly advances the field of DDIM Inversion by addressing the inherent inefficiencies and performance limitations of traditional iterative optimization methods. At the core of our EasyInv is a refined strategy for approximating inversion noise, which is pivotal for enhancing the accuracy and reliability of the inversion process. By prioritizing the initial latent state, which encapsulates rich information about the original images, EasyInv steers clear of the iterative refinement of noise items. Instead, we introduce a methodical aggregation of the latent state from the preceding time step with the current state, effectively increasing the influence of the initial latent state and mitigating the impact of noise. We illustrate that EasyInv is capable of delivering results that are either on par with or exceed those of the conventional DDIM Inversion approach, especially under conditions where the model's precision is limited or computational resources are scarce. Concurrently, our EasyInv offers an approximate threefold enhancement regarding inference efficiency over off-the-shelf iterative optimization techniques. See code at https://github.com/potato-kitty/EasyInv.

## 1. Introduction

Diffusion models are renowned for their ability to generate high-quality images that closely match given prompts. Among the many diffusion models, Stable Diffusion (SD) (Rombach et al., 2022) stands out as one of the most widely utilized models. Another contemporary diffusion

*Equal contribution [1]Key Laboratory of Multimedia Trusted Perception and Efficient Computing, Ministry of Education of China, Xiamen University, 361005, P.R. China [2]Skywork AI, Singapore [3]National University of Singapore. Correspondence to: Rongrong Ji <rrji@xmu.edu.cn>.

*Proceedings of the 42nd International Conference on Machine Learning*, Vancouver, Canada. PMLR 267, 2025. Copyright 2025 by the author(s).

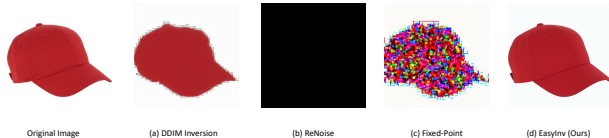

*Figure 1.* Comparison of inversion methods including vanilla DDIM Inversion (Couairon et al., 2023), Fixed-Point Iteration (Pan et al., 2023), ReNoise (Garibi et al., 2024) and our EasyInv.

model gaining popularity is DALL-E 3 (Betker et al., 2023), which offers users access to its API and the ability to interact with it through platforms like ChatGPT (Openai, 2024). These models have transformed the visual arts industry and have attracted substantial attention from the research community. While renowned generative diffusion models have made strides, a prevalent limitation is their reliance on textual prompts for input. This approach becomes restrictive when users seek to iteratively refine an image, as the sole reliance on prompts hinders flexibility. Although solutions such as ObjectAdd (Zhang et al., 2024) and P2P (Hertz et al., 2022) have been proposed to address image editing challenges, they are still confined to the realm of prompted image manipulation. Given that diffusion models generate images from noise inputs, a potential breakthrough lies in identifying the corresponding noise for any given image. This would enable the diffusion model to initiate the generation process from a known starting point, thereby allowing for precise control over the final output. The recent innovation of DDIM Inversion (Couairon et al., 2023) overcomes this by reversing the denoising process to introduce noise. This technique retrieves the initial noise configuration after a series of reference steps, thereby preserving the integrity of the original image while affording the user the ability to manipulate the output by adjusting the denoising parameters. With DDIM inversion, the generative process becomes more adaptable, facilitating the creation and subsequent editing of images with greater precision and control. For example, MasaCtrl (Cao et al., 2023) first transforms a real image into a noise representation and then identifies the arrangement of objects during the denoising phase. Portrait Diffusion (Liu et al., 2023) inverts both the source and target images. It merges their respective $Q$, $K$ and $V$ values for mixups.

Considering the reliance on inversion techniques to preserve the integrity of the input image, the quality of the inversion process is paramount, as it profoundly influences subsequent tasks. As depicted in Figure 1(a), the performance of DDIM Inversion has been found to be less than satisfactory due to the discrepancy between the noise estimated during the inversion process and the noise expected in the sampling process. Consequently, numerous studies have been conducted to enhance its efficacy. In Null-Text inversion (Mokady et al., 2023), researchers observed that using a null prompt as input, the diffusion model could generate optimal results during inversion, suggesting that improvements to inversion might be better achieved in the reconstruction branch. Ju *et al.*'s work (Ju et al., 2023) exemplifies this approach by calculating the distance between latents at the current step and the previous step. PTI (Dong et al., 2023) opts to update the conditional vector in each step to guide the reconstruction branch for improving consistency. ReNoise (Garibi et al., 2024) focuses on refining the inversion process itself. This method iteratively adds and then denoises noise at each time step, using the denoised noise as input for the subsequent iteration. However, as shown in Figure 1(b), it can result in a black image output when dealing with certain special inputs, which will be discussed in detail in Section 4. Pan *et al.* (Pan et al., 2023), while maintaining the iterative updating process, also amalgamated noise from previous steps with the current step's noise. However, this method's performance is limited in less effective models as displayed in Figure 1(c). For instance, it performs well in SD-XL (Podell et al., 2023) but fails to yield proper results in SD-V1-4 (Rombach et al., 2022). We attribute this to their sole focus on optimizing noise; when the noise is highly inaccurate, such simple optimization strategies encounter difficulties. Additionally, the iterative updating of noise is time-consuming, as Pan *et al.*'s method requires multiple model inferences per time step.

In this paper, we conduct an in-depth analysis and recognize that the foundation of any inversion process is the initial latent state derived from a real image. Errors introduced at each step of the inversion process can accumulate, leading to a suboptimal reconstruction. Current methodologies, which focus on optimizing the transition between successive steps, may not be adequate to address this issue holistically. To tackle this, we propose a novel approach that considers the inversion process as a whole, underscoring the significance of the initial latent state throughout the process. Our approach, named EasyInv, incorporates a straightforward mechanism to periodically reinforce the influence of the initial latent state during the inversion. This is realized by blending the current latent state with the previous one at strategically selected intervals, thereby increasing the weight of the initial latent state and diminishing the noise's impact. As a result, EasyInv ensures a reconstructed version that remains closer to the original image, as illustrated in

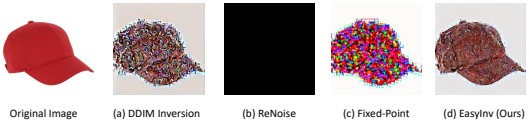

Figure 2. Visualization of latent states at midway denoising steps for various inversion methods. Our EasyInv shows enhanced convergence by closely approximating the original image.

Figure 1(d). Furthermore, by building upon the traditional DDIM Inversion framework (Couairon et al., 2023), EasyInv does not depend on iterative optimization between adjacent steps, thus enhancing computational efficiency. In Figure 2, we present a visualization of the latent states at the midpoint of the total denoising steps for various inversion methods. The outcomes of our EasyInv are more closely aligned with the original image compared to all other methods, demonstrating that EasyInv achieves faster convergence.

## 2. Related Works

**Diffusion Model**. In recent years, there has been significant progress in the field of generative models, with diffusion models emerging as a particularly popular approach. The seminal denoising diffusion probabilistic models (DDPM) (Ho et al., 2020) introduced a practical framework for image generation based on the diffusion process. This method stands out from its predecessors, such as generative adversarial networks (GANs), due to its iterative nature. During the data preparation phase, Gaussian noise is incrementally added to a real image until it transitions into a state that is indistinguishable from raw Gaussian noise. Subsequently, a model can be trained to predict the noise added at each step, enabling users to input any Gaussian noise and obtain a high-quality image as a result. Ho *et al.* (Ho et al., 2020) provided a robust theoretical foundation for their model, which has facilitated further advancements. Generative process in DDPM is both time-consuming and inherently stochastic due to the random noise introduced at each step. To address these limitations, the denoising diffusion implicit models (DDIM) were developed (Song et al., 2020). By reformulating DDPM, DDIM has reduced the amount of random noise added at each step. This reformulation results in a more deterministic denoising process. Furthermore, the absence of random noise allows for the aggregation of several denoising steps, thereby reducing the overall computation time required to generate an image.

**Image Inversion**. Converting a real image into noise is pivotal in real image editing using diffusion models. The precision of this process has a profound impact in the final edit, with the critical element being the accurate identification

of the noise added at each step. Couairon *et al.* (Couairon et al., 2023) swapped the roles of independent and implicit variables within the denoising function of the DDIM model, enabling it to predict the noise that should be introduced to the current latents. However, it is essential to recognize that the denoising step in a diffusion model is inherently an approximation, and when this approximation is utilized inversely, discrepancies between the model's output and the actual noise value are likely to be exacerbated. To address this issue, ReNoise (Garibi et al., 2024) iterates through each noising step multiple times. For each inversion step, they employ an iterative approach to add and subsequently reduce noise, with the noise reduced in the final iteration being carried forward to the subsequent iteration. Pan *et al.* (Pan et al., 2023) offered a theoretical underpinning to the ReNoise. Iterative optimization from ReNoise is classified under the umbrella of fixed-point iteration methods. Building upon Anderson's seminal work (Anderson, 1965), Pan *et al.* have advanced the field by proposing their novel method for optimizing noise during the inversion process.

## 3. Methodology

### 3.1. Preliminaries

**DDIM Inversion**. Let $\mathbf{z}_T$ denote a noise tensor with $\mathbf{z}_T \sim \mathcal{I}(0, \mathbf{I})$. The DDIM (Couairon et al., 2023) leverages a pre-trained neural network $\varepsilon_\theta$ to perform $T$ denoising diffusion steps. Each step aims to estimate the underlying noise and subsequently restore a less noisy version of the tensor, $\mathbf{z}_{t-1}$, from its noisy counterpart $\mathbf{z}_t$ as:

$$
\mathbf{z}_{t-1} = \sqrt{\frac{\alpha_{t-1}}{\alpha_t}} \mathbf{z}_t +
$$
$$
\left( \sqrt{\frac{1}{\alpha_{t-1}} - 1} - \sqrt{\frac{1}{\alpha_t} - 1} \right) \cdot \varepsilon_\theta \big( \mathbf{z}_t, t, \tau_\theta(y) \big), \quad (1)
$$

where $t = T \to 1$, and $\{\alpha_t\}_{t=1}^T$ constitutes a prescribed variance set that guides the diffusion process. Furthermore, $\tau_\theta$ serves as an intermediate representation that encapsulates the textual condition $y$. We denote:

$$
d(\mathbf{z}_t) = \varepsilon_\theta \big( \mathbf{z}_t, t, \tau_\theta(y) \big). \quad (2)
$$

Re-evaluating Equation (1), we derive DDIM Inversion process (Couairon et al., 2023) as presented in Equation (3). In this reformulation, we relocate an approximate $\mathbf{z}_t^*$ to the left-hand side, resulting in the following expression:

$$
\mathbf{z}_t^* = g \left( \varepsilon_\theta \big( \mathbf{z}_{t-1}^*, t-1, \tau_\theta(y) \big) \right)
$$
$$
= \sqrt{\frac{\alpha_t}{\alpha_{t-1}}} \mathbf{z}_{t-1}^* -
$$
$$
\sqrt{\frac{\alpha_t}{\alpha_{t-1}}} \left( \sqrt{\frac{1}{\alpha_{t-1}} - 1} - \sqrt{\frac{1}{\alpha_t} - 1} \right) \cdot \varepsilon_\theta \big( \mathbf{z}_{t-1}^*, t-1, \tau_\theta(y) \big), \quad (3)
$$

**Review**. Given an image $\mathbf{I}^*$, after encoding it into the latent $\mathbf{z}_0^*$, we initiate $T$ inversion steps using Equation (3) to obtain

the noise $\mathbf{z}_T^*$. Starting with $\mathbf{z}_T = \mathbf{z}_T^*$, we proceed with a denoising process in Equation (1) to infer an approximate reconstruction $\mathbf{z}_0$ that resembles the original latent $\mathbf{z}_0^*$. The primary source of error in this reconstruction arises from the difference between the noise predicted during the inversion process $\varepsilon_\theta \big( \mathbf{z}_{t-1}^*, t-1, \tau_\theta(y) \big)$ and the noise expected in the sampling process, $\varepsilon_\theta \big( \mathbf{z}_t, t, \tau_\theta(y) \big)$, denoted as $\varepsilon_t$, at each iterative step. This discrepancy originates from an imprecise approximation of the time step from $t$ to $t-1$. Therefore, reducing the discrepancy between the predicted noises at each step is crucial for achieving an accurate reconstruction, which is essential for the success of subsequent image editing tasks. For simplicity below, we define:

$$
\varepsilon_t^* = \varepsilon_\theta \big( \mathbf{z}_{t-1}^*, t-1, \tau_\theta(y) \big), \ \varepsilon_t = \varepsilon_\theta \big( \mathbf{z}_t, t, \tau_\theta(y) \big). \quad (4)
$$

### 3.2. Fixed-Point Iteration

The vanilla DDIM Inversion method, as discussed, involves an approximation that is not entirely precise for $\varepsilon_t^*$. To address this, researchers have sought to refine a more accurate approximation of $\varepsilon_t^*$, thereby ensuring that the desired conditions are optimally met. This refinement process aims to enhance the precision of the method, leading to more reliable results in the context of the application:

$$
\varepsilon_t^* = \varepsilon_t. \quad (5)
$$

For clarity, let's first restate Equation (3) as follows:

$$
\mathbf{z}_t^* = g(\varepsilon_t^*), \quad (6)
$$

which represents the introduction of adding noise to the latent state $\mathbf{z}_{t-1}^*$. Under the assumption of Equation (5), it should be the case that:

$$
\mathbf{z}_t = \mathbf{z}_t^*. \quad (7)
$$

Subsequently, by employing the noise estimation function from Equation (2), we obtain:

$$
d(\mathbf{z}_t) = d\big(g(\varepsilon_t^*)\big). \quad (8)
$$

Combining $d(\mathbf{z}_t) = \varepsilon_t$ and Equation (5), we obtain:

$$
\varepsilon_t^* = d\big(g(\varepsilon_t^*)\big). \quad (9)
$$

This formulation presents a fixed-point problem, which pertains to a value that remains unchanged under a specific transformation (Bauschke et al., 2011). In the context of functions, a fixed point is an element that is invariant under the application of the function. We seek a $\varepsilon_t^*$ that, when transformed by $g$ and followed by $d$, can map back to itself, signifying an optimal solution as per Equation (5).

Fixed-point iteration is a computational technique designed to identify the fixed points of a function. It functions through an iterative process, as delineated below:

$$(\varepsilon_t^*)^n = d\big(g(\varepsilon_t^*)^{n-1}\big), \tag{10}$$

where $n$ denotes the iteration count. This iterative process can be enhanced through acceleration techniques such as Anderson acceleration (Anderson, 1965). However, calculating a complex $\varepsilon_t^*$ can be quite onerous. An empirical acceleration method proposed (Pan et al., 2023) introduces a refinement for $\varepsilon_t^*$ by setting: $(\varepsilon_t^*)^n = \varepsilon_\theta((\mathbf{z}_t^*)^n, t-1, \tau_\theta(y))$ and $(\varepsilon_t^*)^{n-1} = \varepsilon_\theta\big((\mathbf{z}_t^*)^{n-1}, t-1, \tau_\theta(y)\big)$. They finally reach:

$$(\mathbf{z}_t^*)^{n+1} = g(0.5 \cdot (\varepsilon_t^*)^n + 0.5 \cdot (\varepsilon_t^*)^{n-1}), \tag{11}$$

where the term $0.5 \cdot (\varepsilon_t^*)^n + 0.5 \cdot (\varepsilon_t^*)^{n-1}$ represents the refinement technique for $\varepsilon_t^*$ as suggested by Pan *et al.* If we were to apply the function $d$ to both sides of Equation (11), it would align perfectly with the form of Equation (10). Their experiments have demonstrated that this approach is more effective than both Anderson's method (Anderson, 1965) and other techniques in inversion tasks.

Despite the progress made, this paper acknowledges inherent limitations in the practical implementation of the inversion technique. (1) Inversion Efficiency: While the method outlined in Equation (11) has shown improvements over traditional fixed-point iteration, it still relies on iterative optimization. The need for multiple forward passes through the diffusion model is computationally demanding and can result in inefficiencies in downstream applications. (2) Inversion Performance: The theoretical improvements presented assume that $\varepsilon_t^* = \varepsilon_t$. However, iterative optimization does not guarantee the exact fulfillment of Equation (7) for every time step $t$. Therefore, while the method may theoretically offer superior performance, cumulative errors can sometimes lead to practical outcomes that are less satisfactory than those achieved with the standard DDIM Inversion method, as shown in Figure 1.

### 3.3. Kalman Filter

The Kalman filter is designed to estimate the state of a dynamic system from noisy measurements. It assumes two states of the system: the predicted state, $x_k$, which is calculated using the process function, and the observed state, $y_k$, which represents the measured state of the system. For instance, the position of an object could be predicted using its velocity and time, while its position might be observed via a radar. Here, $k$ denotes the time-step. Ideally, $x_k$ and $y_k$ should be identical; however, in practice, both are prone to inaccuracies. The Kalman filter aims to provide an optimized estimate by combining both values. The relevant equations are as follows:

$$x_k = Ax_{k-1} + Bu_k + w_{k-1}. \tag{12}$$

$$y_k = Hx_k + v_k. \tag{13}$$

In these equations, $x_k$ represents the true value at time-step $k$, which we aim to estimate; $u_{k-1}$ is the control input at time step $k-1$, such as acceleration; $w_{k-1}$ and $v_k$ are random noise or errors introduced during calculation and measurement, thus $y_k$ represents the measurement value; $A$ and $B$ are weight matrices; and $H$ is the transformation matrix. When $x_k$ and $y_k$ are of the same type, the matrix $H$ can be omitted. In practice the value of $w_{k-1}$ is unknown, otherwise the precise results would be calculated without the needs of Kalman filter. Instead, we have the predicted value $\bar{x}_k$, which is $\bar{x}_k = Ax_{k-1} + Bu_k$. The key idea of the Kalman filter is to fuse these $\bar{x}_k$ and $y_k$ in order to obtain a more accurate estimation of the system state. The fusion process is typically represented as:

$$\tilde{x}_k = \alpha\bar{x}_k + \beta y^{measure}, \tag{14}$$

where $\alpha + \beta = I$ and $y^{measure} = H^{-1}y_k$. We can rewrite the fusion equation as:

$$\tilde{x}_k = (1-\beta)\bar{x}_k + \beta H^{-1}y_k = \bar{x}_k + \beta(H^{-1}y_k - \bar{x}_k), \tag{15}$$

When we set $\beta = K \cdot H$ the objective function would be:

$$\tilde{x}_k = \bar{x}_k + K(y_k - H\bar{x}_k). \tag{16}$$

Here, $K$ is the Kalman gain. Additionally, $\bar{x}_k = A\tilde{x}_{k-1} + Bu_k$ is the predicted estimate of the true state $x_k$, based on the previous estimate $\tilde{x}_{k-1}$.

### 3.4. EasyInv

To facilitate our subsequent analysis, we introduce the notation $\bar{\alpha}_t$ to represent $\sqrt{\frac{\alpha_t}{\alpha_{t-1}}}$ and $\bar{\beta}_t$ to denote $\sqrt{\frac{\alpha_t}{\alpha_{t-1}}}\left(\sqrt{\frac{1}{\alpha_{t-1}}-1} - \sqrt{\frac{1}{\alpha_t}-1}\right)$. With these notations, we can reframe Equation (3) as follow:

$$\mathbf{z}_t^* = \bar{\alpha}_t\mathbf{z}_{t-1}^* + \bar{\beta}_t\varepsilon_t^*. \tag{17}$$

Similarly, we can express the form of $\mathbf{z}_{t-1}^*$ as:

$$\mathbf{z}_{t-1}^* = \bar{\alpha}_{t-1}\mathbf{z}_{t-2}^* + \bar{\beta}_{t-1}\varepsilon_{t-1}^*. \tag{18}$$

By combining these two formulas, we derive:

$$\mathbf{z}_t^* = \bar{\alpha}_t\bar{\alpha}_{t-1}\mathbf{z}_{t-2}^* + \bar{\alpha}_t\bar{\beta}_{t-1}\varepsilon_{t-1}^* + \bar{\beta}_t\varepsilon_t^*. \tag{19}$$

This can be further generalized to:

$$\mathbf{z}_t^* = (\prod_{i=1}^{t}\bar{\alpha}_i)\mathbf{z}_0^* + \sum_{i=1}^{t}(\bar{\beta}_i\prod_{j=i+1}^{t}\bar{\alpha}_j)\varepsilon_i^*. \tag{20}$$

From Equation (20), it is evident that $\mathbf{z}_t^*$ is a weighted sum of $\mathbf{z}_0$ and a series of noise terms $\varepsilon_i^*$. The denoising process of Equation (1) aims to iteratively reduce the impact of these noise terms. In prior research, the crux of inversion is to introduce the appropriate noise $\varepsilon_i^*$ at each step to identify a suitable $\mathbf{z}_t^*$. This allows the model to obtain $\mathbf{z}_0$ as the final output after the denoising process. However, iteratively updating $\varepsilon_i^*$ can be time-consuming, and when the model lacks high precision, achieving satisfactory results within a reasonable number of iterations may be challenging.

To address this, we propose an alternative perspective. During inversion, rather than searching for better noise, we aggregate the latent state from the last time step $\mathbf{z}_{\bar{t}-1}^*$ with the current latent state $\mathbf{z}_{\bar{t}}^*$ at specific time steps $\bar{t}$, as illustrated in the following formula:

$$\mathbf{z}_{\bar{t}}^* = \eta\mathbf{z}_{\bar{t}}^* + (1-\eta)\mathbf{z}_{\bar{t}-1}^*, \qquad (21)$$

where $\eta$ is a trade-off parameter. The selection of $\bar{t}$ and $\eta$ will be discussed in Section 4.1. Considering Equation (16), when applying it to solve the inversion problem, we obtain the following equation:

$$\mathbf{z}_{\bar{t}}^* = K\mathbf{z}_{\bar{t}}^* + (1-K)y_k, \qquad (22)$$

where $H$ is ignored since $y_k$ and $\bar{x}_k$ show be same kind of data. $\mathbf{z}_{\bar{t}}^*$ represents the estimated value $\tilde{x}_k$. Next, we replace the measured value $y_k$ with $\mathbf{z}_{\bar{t}-1}^*$, because while $z_t$ cannot be directly measured during the inversion process at timestep $t-1$, the difference $v_t = z_t - z_{t-1}$ is assumed to be Gaussian noise, in accordance with the basic principles of diffusion models. Thus, $z_{t-1}$ can be treated as the measured value $x_k$ in Equation (13) for the inversion problem. The equation then becomes:

$$\mathbf{z}_{\bar{t}}^* = K\mathbf{z}_{\bar{t}}^* + (1-K)\mathbf{z}_{\bar{t}-1}^*. \qquad (23)$$

Typically, determining the Kalman Gain $K$ is the main goal of the Kalman filter algorithm, which can be computationally expensive. However, the approach described in Equation (21) can be viewed as a simplified version of the Kalman filter, where $K$ is treated as a constant value, determined empirically. With this method, the added noise for inverting $\mathbf{z}_{\bar{t}-1}^*$ to $\mathbf{z}_{\bar{t}}^*$ would be:

$$\mathbf{z}_{\bar{t}}^* - \mathbf{z}_{\bar{t}-1}^* = \eta(\bar{\mathbf{z}}_{\bar{t}}^* - \mathbf{z}_{\bar{t}-1}^*). \qquad (24)$$

Here, $\bar{\mathbf{z}}_{\bar{t}}^*$ represents the original $\mathbf{z}_{\bar{t}}^*$ on the right-hand side of Equation (21), generated via DDIM inversion in our case. As indicated in Equation (24), the distribution transition from $\mathbf{z}_{\bar{t}-1}^*$ to $\mathbf{z}_{\bar{t}}^*$ in our method mirrors the distribution pattern of DDIM inversion, maintaining the same variance. Due to $\eta < 1$, the average value is reduced. In other words, our method preserves the noise pattern of the original method but applies a rescaling factor.

Algorithm 1 illustrates the approximate operation of our method in conjunction with an inversion framework.

---

**Algorithm 1** Add EasyInv to an existing inversion method

**Require:** A inversion algorithm $Inv()$, total inversion steps $T$, latent $z$, chosen steps $\bar{t}$, empirical parameter $\eta$
1: **for** $t$ in $T$ **do**
2:    $z_{t+1} = Inv(z_t, t)$
3:    **if** $t$ in $\bar{t}$ **then**
4:       $z_{t+1} = \eta * z_{t+1} + (1-\eta) * z_t$
5:    **end if**
6: **end for**
**Output:** inverted latent $z_T$

---

## 4. Experimentation

In Table 1 and Table 2, we compare our EasyInv over other inversion methods, using SD V1.4 on one NVIDIA GTX 3090 GPU. For Fixed-Point Iteration (Pan et al., 2023), we re-implemented it using settings from the paper. We set the data type of all methods to float16 by default to improve efficiency. The inversion and denoising steps are $T = 50$, except for Fixed-Point Iteration, which recommends $T = 20$. For our EasyInv, we set $0.05 \cdot T < \bar{t} < 0.25 \cdot T$ and $\eta = 0.5$. In Table 3 we compare the performance of different down-string tasks when using different inversion methods, the dataset and codes we used in this experiment are from PNPinversion (Ju et al., 2024).

We use three major qualitative metrics: LPIPS index (Zhang et al., 2018), SSIM (Wang et al., 2004), and PSNR with the inference time. We sample 2,298 images from the COCO 2017 test and validation sets (Lin et al., 2014).

### 4.1. Quantitative Results

Table 1 presents the quantitative results of different methods. EasyInv achieves a competitive LPIPS score of 0.321, better than ReNoise (0.316) and Fixed-Point Iteration (0.373), indicating closer perceptual similarity to the original image. For SSIM, EasyInv achieves the highest score of 0.646, showing superior structural similarity crucial for maintaining

*Table 1.* A comparative result of quantitative outcomes.

| | LPIPS ($\downarrow$) | SSIM ($\uparrow$) | PSNR ($\uparrow$) | Time ($\downarrow$) |
|---|---|---|---|---|
| DDIM Inversion | 0.328 | 0.621 | 29.717 | **5s** |
| ReNoise | **0.316** | 0.641 | **31.025** | 16s |
| Fixed-Point Iteration | 0.373 | 0.563 | 29.107 | 14s |
| EasyInv (Ours) | 0.321 | **0.646** | 30.189 | **5s** |

*Table 2.* A comparative result of half- and full-precision EasyInv.

| | LPIPS ($\downarrow$) | SSIM ($\uparrow$) | PSNR ($\uparrow$) | Time ($\downarrow$) |
|---|---|---|---|---|
| Full Precision | 0.321 | 0.646 | 30.184 | 9s |
| Half Precision | 0.321 | 0.646 | **30.189** | **5s** |

*Table 3.* Performance of downstream methods. Results except "Ours+DirectInv" are from PNPinversion (Ju et al., 2024). DDIM inversion (Couairon et al., 2023) for DDIM, Null-text inversion (Mokady et al., 2023) for NT, Negative-Prompt Inversion (Miyake et al., 2023) for NP, StyleDiffusion (Wang et al., 2023) for StyleD, PNPinversion (Ju et al., 2024) for DirectInv, Prompt-to-Prompt (Hertz et al., 2022) for P2P, MasaCtrl (Cao et al., 2023) for MasaCtrl, Pix2Pix-Zero (Parmar et al., 2023) for P2P-Zero and Plug-and-Play (Tumanyan et al., 2023) for PnP.

† averaged results on A800 and RTX3090 since different environment leads to slightly different performance.

∗ uses Stable Diffusion v1.5 as base model (others all use Stable Diffusion v1.4).

| Method | | Structure | Background Preservation | | | | CLIP Similarity | |
|---|---|---|---|---|---|---|---|---|
| Inverse | Editing | Distance$_{\times 10^3}$ ↓ | PSNR ↑ | LPIPS$_{\times 10^3}$ ↓ | MSE$_{\times 10^4}$ ↓ | SSIM$_{\times 10^2}$ ↑ | Whole ↑ | Edited ↑ |
| DDIM | P2P | 69.43 | 17.87 | 208.80 | 219.88 | 71.14 | 25.01 | 22.44 |
| NT† | P2P | 13.44 | 27.03 | 60.67 | 35.86 | 84.11 | 24.75 | 21.86 |
| NP | P2P | 16.17 | 26.21 | 69.01 | 39.73 | 83.40 | 24.61 | 21.87 |
| StyleD | P2P | 11.65 | 26.05 | 66.10 | 38.63 | 83.42 | 24.78 | 21.72 |
| DirectInv | P2P | 11.65 | 27.22 | 54.55 | 32.86 | 84.76 | 25.02 | 22.10 |
| **Ours+DirectInv** | P2P | **11.58** | **27.30** | **53.52** | **32.37** | **84.80** | 24.97 | 22.00 |
| DDIM | MasaCtrl | 28.38 | 22.17 | 106.62 | 86.97 | 79.67 | 23.96 | 21.16 |
| DirectInv | MasaCtrl | 24.70 | 22.64 | 87.94 | 81.09 | 81.33 | 24.38 | 21.35 |
| **Ours+DirectInv** | MasaCtrl | **23.82** | **22.72** | **87.65** | **79.73** | **81.82** | 24.36 | 21.32 |
| DDIM | P2P-Zero | 61.68 | 20.44 | 172.22 | 144.12 | 74.67 | 22.80 | 20.54 |
| DirectInv | P2P-Zero | 49.22 | 21.53 | 138.98 | 127.32 | 77.05 | 23.31 | 21.05 |
| **Ours+DirectInv** | P2P-Zero | **48.02** | **21.54** | **136.78** | **124.00** | **77.71** | **23.40** | 20.93 |
| DDIM | PnP∗ | 28.22 | 22.28 | 113.46 | 83.64 | 79.05 | 25.41 | 22.55 |
| DirectInv | PnP∗ | 24.29 | 22.46 | 106.06 | 80.45 | 79.68 | 25.41 | 22.62 |
| **Ours+DirectInv** | PnP∗ | **22.88** | **22.56** | **102.34** | **78.57** | **80.27** | 25.38 | 22.53 |

image coherence. For PSNR, EasyInv scores 30.189, close to ReNoise's highest score of 31.025, indicating high image fidelity. EasyInv completes the inversion process in the fastest time of 5 seconds, matching DDIM Inversion (Couairon et al., 2023), and quicker than ReNoise (16 seconds) and Fixed-Point Iteration (14 seconds), highlighting its efficiency without compromising on quality. EasyInv performs strongly across all metrics, with the highest SSIM score indicating effective preservation of image structure. Its efficient inversion makes it suitable for real-world applications where both quality and speed are crucial.

Table 2 compares EasyInv's performance in half-precision (float16) and full-precision (float32) formats. Both achieve the same LPIPS score of 0.321, indicating consistent perceptual similarity to the original image. Similarly, both achieve an SSIM score of 0.646, showing preserved structural integrity with high fidelity. For PSNR, half precision slightly outperforms full precision with scores of 30.189 and 30.184. This slight advantage in PSNR for half precision is noteworthy given its well reduced computation time. The most significant difference is observed in the time metric, where half precision completes the inversion process in 5 seconds, approximately 44% faster than full precision, which takes 9 seconds. This efficiency gain highlights EasyInv's excep-

tional optimization for half precision, offering faster speeds and reduced resources without compromising output quality.

In Table 3, we compare the performance of several downstream tasks using our method and other inversion techniques. Most of the results in this table are from PNPInversion (Ju et al., 2024), where a dataset for 9 different image editing tasks is introduced, along with the corresponding code for both editing and evaluation. We used these codes and dataset for this experiment. The only modification we made was to incorporate our method into DirectInv, the inversion method proposed in their work (Ju et al., 2024), and its results are indicate as 'ours+DirectInv' in Table 3. As we pointed out, our method is able to combined with most existing inversion algorithm. The results in Table 3 show the our advantage. By adding our method, the performance of DirectInv improves in 5 out of 7 metrics across all editing tasks, with minimal changes in the remaining 2 metrics.

### 4.2. Qualitative Results

We visually evaluate all methods using SD-V1-4. Figure 3 displays the results using images sourced from the internet. These images also feature large areas of white color. ReNoise struggles with images containing significant white areas, resulting in black images. Fixed-Point Iteration and

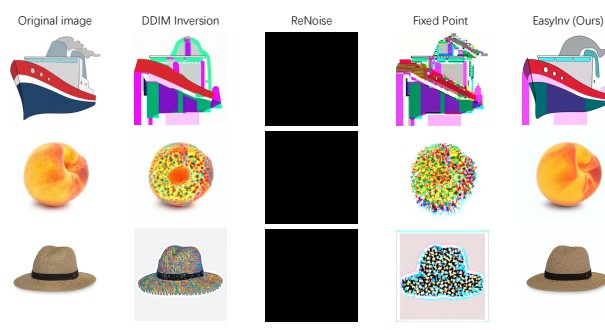

Figure 3. A visual assessment of various inversion techniques.

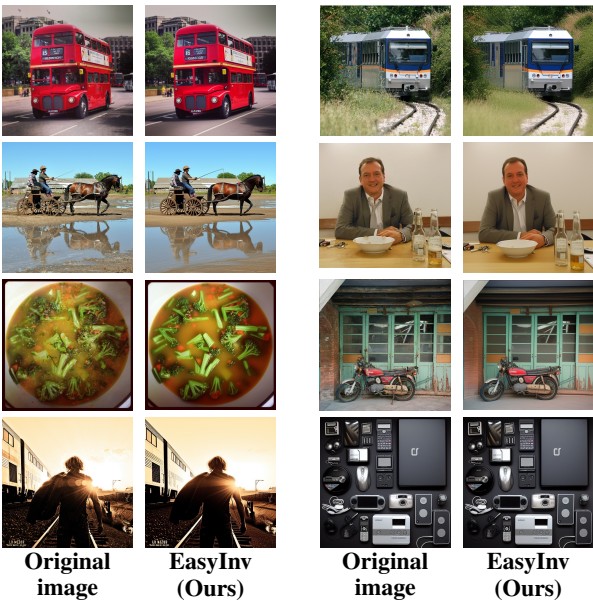

| Original image | EasyInv (Ours) | Original image | EasyInv (Ours) |
| --- | --- | --- | --- |

Figure 4. More visual results of our EasyInv.

DDIM Inversion also fail to generate satisfactory results in such cases, suggesting these images pose challenges for inversion methods. Our method, shown in the figure, effectively addresses these challenges, demonstrating robustness and enhancing performance in handling special scenarios. These findings underscore the efficacy of our approach, particularly in addressing challenging cases that are less common in the COCO dataset. For better illustrations, we show more visual examples in Figure 4.

### 4.3. Ablation Studies

To clarify our experimental choices, in Table 4 we present ablation studies on the parameter $\eta$, conducted on the same dataset as in Table 3. Note that $\eta = 1$ would correspond to a standard DDIM inversion and $\eta = 0$ would bypass the inversion operation. Therefore, we exclude them. Our findings indicate that $\eta = 0.5$ yields the best overall performance.

*Table 4.* Ablation Experiments on $\eta$.

| Setting | | Structure | Background Preservation | | | | CLIP Similarity | |
| --- | --- | --- | --- | --- | --- | --- | --- | --- |
| $\eta$ | Editing | Distance$_{\times 10^3}$ ↓ | PSNR ↑ | LPIPS$_{\times 10^3}$ ↓ | MSE$_{\times 10^4}$ ↓ | SSIM$_{\times 10^2}$ ↑ | Whole ↑ | Edited ↑ |
| 0.2 | PnP* | 25.90 | **23.29** | 113.86 | **66.11** | 80.13 | 24.66 | 21.60 |
| 0.4 | PnP* | 24.44 | **23.29** | 107.18 | 66.32 | 80.69 | 24.82 | 21.76 |
| 0.5 | PnP* | **22.88** | 22.56 | 102.34 | 78.57 | 80.27 | **25.38** | **22.53** |
| 0.6 | PnP* | 23.36 | 23.21 | 101.37 | 67.96 | **81.07** | 25.02 | 22.00 |
| 0.8 | PnP* | 22.89 | 22.92 | **98.44** | 72.30 | 80.94 | 25.30 | 22.29 |

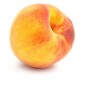 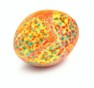 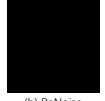 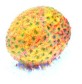 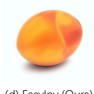

| Original Image | (a) DDIM Inversion | (b) ReNoise | (c) Fixed-Point | (d) EasyInv (Ours) |
| --- | --- | --- | --- | --- |

Figure 5. Results of MasaCtrl (Cao et al., 2023) with prompt "A football", using inverted latent by different methods as input.

### 4.4. Downstream Image Editing

To showcase the practical utility of our EasyInv, we have employed various inversion techniques within the realm of consistent image synthesis and editing. We have seamlessly integrated these inversion methods into MasaCtrl (Cao et al., 2023), a widely-adopted image editing approach that extracts correlated local content and textures from source images to ensure consistency. For demonstrative purposes, we present an image of a "peach" alongside the prompt "A football." The impact of inversion quality is depicted in Figure 5. It is clear that our inversion method demonstrates superior performance in both texture and shape. We also apply our EasyInv to two additional downstream tasks: DiffusionTrend (Zhan et al., 2024) for virtual try-on and UniVST (Song et al., 2024) for video style transferring. In Figure 6, using our method results in better consistency of the cloth compared to DDIM and results in Figure 7 show that DDIM inversion leads to chaotic outcomes, whereas our method yields much clearer results.

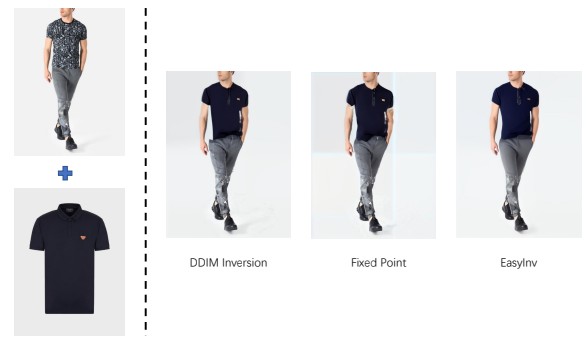

| DDIM Inversion | Fixed Point | EasyInv |
| --- | --- | --- |

Figure 6. Comparison from DiffusionTrend (Zhan et al., 2024).

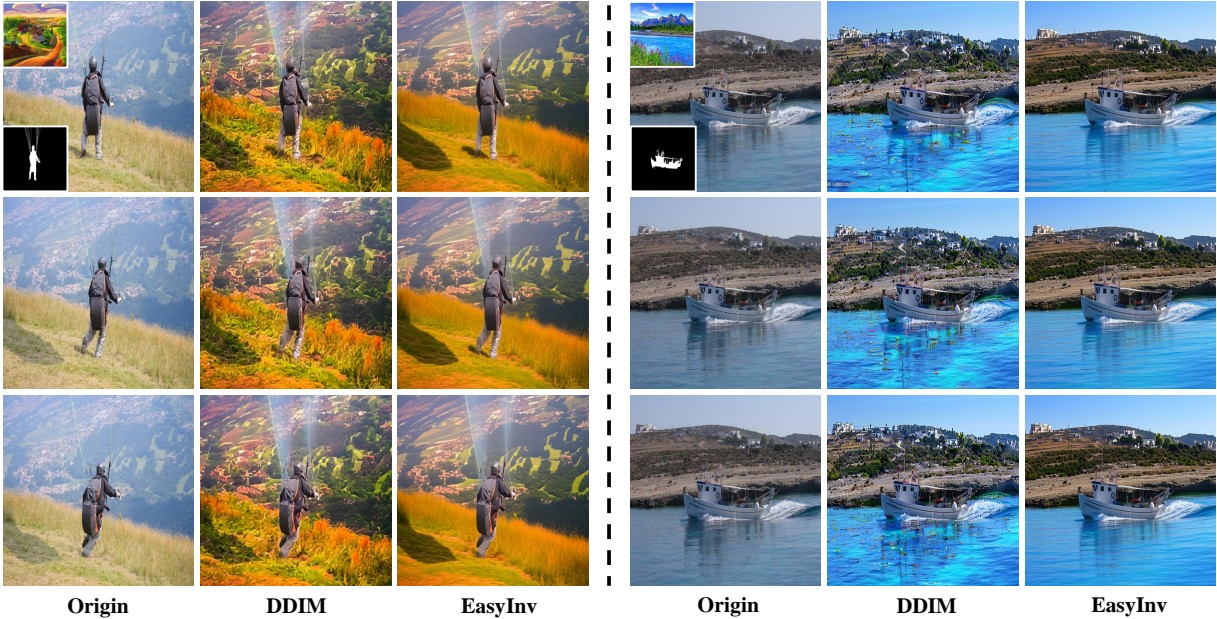

*Figure 7.* Comparison from UniVST ([Song et al., 2024](#)).

### 4.5. Limitations

One potential risk is the phenomenon known as "over-denoising," which occurs when there is a disproportionate focus on achieving a pristine final-step latent state. This may result in overly smooth image outputs, as exemplified by the "peach" figure in Figure 3. In most real-world image editing tasks, this is not a typical issue, as these tasks often involve style migration, which inherently alters the details of the original image. However, in specific applications, such as using diffusion models for creating advertisements, this could pose a challenge. Nonetheless, our experimental results highlight that the method's two key benefits outweigh this minor shortcoming. Firstly, it is capable of delivering satisfactory outcomes even with models that may underperform relative to other methods. Secondly, it enhances inversion efficiency by reverting to the original DDIM Inversion baseline ([Couairon et al., 2023](#)), thereby eliminating the necessity for iterative optimizations. This strategy not only simplifies the process but also ensures the maintenance of high-quality outputs, marking it as a noteworthy advancement over current methodologies.

In conclusion, our research has made significant strides with the introduction of EasyInv. As we look ahead, our commitment to advancing this technology remains unwavering. Our future research agenda will be focused on the persistent enhancement and optimization of the techniques. This will be done with the ultimate goal of ensuring that our method is not only robust and efficient but also highly adaptable to the diverse and ever-evolving needs of industrial applications.

### 5. Conclusion

Our EasyInv addresses the inefficiencies and performance limitations in traditional iterative optimization methods. By emphasizing the importance of the initial latent state and introducing a refined strategy for approximating inversion noise, EasyInv enhances both the accuracy efficiency of the inversion process. Our method reinforces the initial latent state's influence, mitigating the impact of noise and ensuring a closer reconstruction to the original image. This approach not only matches but often surpasses the performance of existing DDIM Inversion methods, especially in scenarios with limited model precision or computational resources. EasyInv also shows a remarkable improvement in inference efficiency, achieving approximately three times faster processing than standard iterative techniques. Through extensive evaluations, we have shown that EasyInv delivers high-quality results, making it a robust and efficient solution for image inversion tasks. The simplicity and effectiveness of EasyInv underscore its potential for broader applications.

### Acknowledgments

This work was supported by the National Science Fund for Distinguished Young Scholars (No.62025603), the National Natural Science Foundation of China (No. U21B2037, No. U22B2051, No. U23A20383, No. 62176222, No. 62176223, No. 62176226, No. 62072386, No. 62072387, No. 62072389, No. 62002305 and No. 62272401), and the Natural Science Foundation of Fujian Province of China (No. 2021J06003, No.2022J06001).

## Impact Statement

This paper presents work whose goal is to advance the field of Machine Learning. There are many potential societal consequences of our work, none which we feel must be specifically highlighted here.

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
