# OpenReview forum: "EasyInv: Toward Fast and Better DDIM Inversion"
_ICML.cc/2025/Conference — ICML 2025 poster_

### Official Review · Reviewer_md9u · 2025-02-16

**Overall Recommendation:** 2

**Summary:**

This paper introduces ​**EasyInv**, an additional mixture operation integrated into the diffusion model-based image editing pipeline, aiming to ​**enhance inversion accuracy** and ​**improve computational efficiency**.

## update after rebuttal
1. I acknowledge the approximation of the Kalman Filter and additional experiments, thus raising the score to 2.5.
2. However, the inversion visual effect in Figure 2 is not that satisfactory, so I cannot give a higher score.
3. And I still doubt about analogizing learned score as Kalman noise; however, if it works in practice, it's fine, and an intuitive starting point may deviate from the underlying true mechanism.

**Claims And Evidence:**

**My concerns about the intuition behind EasyInv's Kalman filter analogy:**

1. ​**Over-Simplification of the Kalman Gain**
   - The original Kalman gain matrix dynamically adapts to uncertainty estimates, but in EasyInv, it is reduced to a ​**constant value**.
   - This simplification strips away the core adaptive mechanism of Kalman filters, raising questions about whether the analogy holds theoretical rigor.

2. ​**Questionable Noise Equivalence Assumption**
   - The derivation from Equations (12)-(13) to (14) implicitly assumes ​$ v_k = w_{k-1} $, where:
     - $ v_k $: Dynamic noise (process uncertainty),
     - $ w_{k-1} $: Measurement noise (observation uncertainty).
   - These two terms are fundamentally ​**independent** in Kalman filtering theory. Asserting their equivalence lacks justification and introduces a contradiction: combining two independent noise sources should not yield cancellation ($ v_k - w_{k-1} = 0 $).

3. ​**Misleading Analogy with PF-ODE Inversion**
   - The term $ \epsilon^* $ in PF-ODE inversion represents ​**scaled diffused data scores**, not random noise.
   - Unlike Kalman filter noise (stochastic by nature), $ \epsilon^* $ encodes meaningful gradient directions derived from data. Equating them conflates ​**stochastic control theory** with ​**deterministic score-based diffusion processes**, weakening the theoretical foundation.

**Essential References Not Discussed:**

**The literature review appears to be somewhat limited.**
- A range of ​**exact inversion sampler techniques**, such as EDICT [1], BDIA [2], and O-BELM [3], have not been discussed or included in the experiments.
- Incorporating these methods into the analysis could provide a more comprehensive understanding of the landscape and strengthen the paper's contributions.

**References:**
[1] Wallace, Bram, Akash Gokul, and Nikhil Naik. "Edict: Exact diffusion inversion via coupled transformations." *Proceedings of the IEEE/CVF Conference on Computer Vision and Pattern Recognition*. 2023.

[2] Zhang, Guoqiang, Jonathan P. Lewis, and W. Bastiaan Kleijn. "Exact diffusion inversion via bidirectional integration approximation." *European Conference on Computer Vision*. 2024.

[3] Wang, Fangyikang, et al. "BELM: Bidirectional Explicit Linear Multi-step Sampler for Exact Inversion in Diffusion Models." *NeurIPS*. 2024.

**Experimental Designs Or Analyses:**

**The impact of the two hyperparameters on performance remains unclear.**
- The choice of $\eta = 0.5$ appears to be ​**heuristic**, lacking a theoretical or empirical justification.
- When applying this method to ​**different tasks**, determining the optimal hyperparameter values becomes a critical challenge.
- Conducting ​**ablation studies** on these hyperparameters could provide valuable insights into their influence and help establish guidelines for task-specific tuning.

**Methods And Evaluation Criteria:**

The evaluation of  EasyInv on image editing and PIEbench is reasonable.

**Other Comments Or Suggestions:**

no

**Other Strengths And Weaknesses:**

The idea is expressed clearly, which is easy to follow.

A lot of visual comparisons are provided.

**Questions For Authors:**

no

**Relation To Broader Scientific Literature:**

This work mainly leverages some findings in the control theory to devise enhanced methods in diffusion model-based inversion.

**Theoretical Claims:**

**The paper lacks theoretical proofs.**
- It is unclear whether the ​**marginal distribution** of the inverted distribution converges to the ​**diffused distribution**.
- EasyInv is presented as a solver for the reverse PF-ODE with ​**order of accuracy 0**. However, no proof is provided to demonstrate that the resulting random variable $ z_t $ of the EasyInv distribution adheres ​**distributionally** to the diffused distribution $ p_t $.
- A formal proof of this adherence would significantly strengthen the theoretical foundation of the proposed method.

---

> ### Author Rebuttal · Authors · 2025-03-28
>
> Thank you for reviewing our paper. The novelty of our work is well recognized by the other two reviewers. However, we received the lowest score from Reviewer md9u. We appreciate your feedback and hope our explanations below address your concerns, leading to a score reconsideration.
>
>
> **1. Over-Simplification of the Kalman Gain and Questionable Noise Equivalence Assumption**
>
> Section 3.3 provides essential background on the Kalman filter, with Equations (12)--(14) summarizing standard formulations consistent with Section 7 of [a]. Notably, the cited reference explicitly states that for specific applications, the Kalman gain $K$ in Equation (14) can be simplified to a blending factor within $[0,1]$, a strategy directly adopted in our methodology. Furthermore, Equation (6) of KLDD [b] demonstrates that the Kalman gain can evolve into a time-varying constant at each timestep, aligning with our approach.
>
> [a] Kalman and bayesian filters in python.
>
> [b] Kldd: Kalman filter based linear deformable diffusion model in retinal image segmentation.
>
> **2. Misleading Analogy with PF-ODE Inversion**
>
> Regarding $\varepsilon^\ast$, while theoretically derived through score-matching optimization, it is important to clarify that this term is not simply a scaled score. Instead, as shown in Equation (15), it is scaled by $\bar{\beta}_t$. In practice, during SDv1.5 training, the model learns to predict the Gaussian noise added to original images [c]. Given its strong noise prediction capabilities, assuming $\varepsilon^\ast$ follows Gaussian characteristics is reasonable.
>
> [c] High-resolution image synthesis with latent diffusion models.
>
>
> **3.Distribution between $z_t$ and $p_t$**
>
> Our formulation preserves the latent variable $z$'s distributional consistency. Equations (18)--(19) show that at the final inversion step ($\bar{t}=T$), $z_T$ emerges as a convex combination of $\eta z_T$ and $(1-\eta)z_{T-1}$. Since ${z_T}$ and ${z}_{T-1}$ are close to the sum of a series of $\varepsilon_i^\ast$ (Gaussian-distributed), their linear combination retains Gaussian properties. To validate this, we computed the KL divergence between inversed latent representations and random input noise in SDv1.5, yielding an average KL divergence of 210.69049 on 298 samples. For comparison, the average KL divergence across 298 randomly generated noise pairs was 247.50026. These results suggest our method effectively preserves latent space distribution.
>
> **4. Studies on Parameter $\eta$**
>
> To clarify our experimental choices, we conduct additional ablation studies on parameter $\eta$ using the same dataset as Table 3. Since $\eta = 1$ corresponds to standard DDIM inversion and $\eta = 0$ bypasses inversion, these cases were excluded. Our results show that $\eta = 0.5$ achieves the best performance.
>
> | $\eta$ | Editing Method | Distance $\(\times 10^2\)\(\downarrow\)$ | PSNR $\(\uparrow\)$ | LPIPS $\(\times 10^3\)\(\downarrow\)$ | MSE $\(\times 10^3\)\(\downarrow\)$ | SSIM $\(\times 10^2\)\(\uparrow\)$ | Whole CLIP Similarity$\(\uparrow\)$ | Edited CLIP Similarity$\(\uparrow\)$ |
> |:-------:|:------:|:---------------------------------------:|:-----------------:|:------------------------------------:|:---------------------------------:|:--------------------------------:|:-------------------:|:--------------------:|
> | η=0.2   | PnP*      | 25.90          | **23.29**  | 113.86       | **66.11**     | 80.13      | 24.66   | 21.60    |
> | η=0.4   | PnP*      | 24.44          | **23.29**  | 107.18       | 66.32     | 80.69      | 24.82   | 21.76    |
> | η=0.5   | PnP*      | **22.88**          | 22.56  | 102.34       | 78.57     | 80.27      | **25.38**   | **22.53**    |
> | η=0.6   | PnP*      | 23.36          | 23.21  | 101.37       | 67.96     | **81.07**      | 25.02   | 22.00    |
> | η=0.8   | PnP*      | 22.89          | 22.92  | **98.44**        | 72.30     | 80.94      | 25.30   | 22.29    |
>
> **5. Extra  literature reviews**
>
> We appreciate the suggestion to broaden our literature review. The three cited methods—EDICT, BDIA, and O-BELM—are classified as exact inversion techniques.
>
> * **EDICT** extends affine coupling mechanisms with alternating dual-state refinement through reciprocal updates between primary and auxiliary diffusion variables.
>
> * **BDIA** enhances this approach with symmetric bidirectional state integration, ensuring provable invertibility but introducing trade-off parameters that may affect robustness across frameworks.
>
> * **O-BELM** synthesizes concepts from both methods, employing a bidirectional explicit architecture while replacing heuristic hyperparameters with analytically derived coefficients optimized via local truncation error minimization, improving numerical stability.
>
> While these methods highlight system reversibility, their empirical updates introduce trade-offs. We focus on fixed-point iteration but acknowledge that incorporating these methods could enhance our analysis. We will include them in the final version.

---

> > ### Comment · Reviewer_md9u · 2025-04-02
> >
> > Thanks for your rebuttal.
> >
> > The Extra literature reviews and ablation study are greatly appreciated. Though the ablation study shows that the $\eta = 0.5$ cannot achieve the best across every metric, the choice of $\eta$ on other datasets is also unclear.
> > Due to the practical contribution of this paper, I will raise my score to 2.
> >
> > However, my concerns about Kalman's theory remain; the issue of $v_k = w_{k-1}$ is not straightforwardly answered.
> > Simplification of the Kalman gain to a constant is ok for me now, because other literature also did this.
> >
> > > It is important to clarify that this term is not simply a scaled score. Instead, as shown in Equation (15), it is scaled by $\bar{\beta}_t$.
> >
> > A score scaled by $\bar{\beta}_t$ is indeed a scaled score.
> >
> >
> > update: please just tell me how to derive eq 14 via eq 12 and eq 13.

---

> > > ### Author Response · Authors · 2025-04-02
> > >
> > > Dear Reviewer,
> > >
> > > We would like to clarify that we did not implicitly assume that $v_k\ =\ w_{k-1}$. In formula (12) of our paper, $x_k$ represents the system state. Because our system may not operate exactly as assumed, $w_{k-1}$ is added in the end of that formula as representation of potential bias. The $y_k$ in formula (13) is the measured state of the system, and $v_k$ is introduced as the measurement bias for the same reason. Regarding formula (14), we have to remind reviewers that it is $\bar{x}\_k$ here in the right-hand side of this formula, not $x_k$. As mentioned, $x_k$ denotes the theoretical state of the system, but in practice the value of $w\_{k-1}$ is unknown, otherwise the precise results would be calculated without the needs of Kalman filter. Instead, we have the predicted value ${\bar{x}}\_k$, which is ${\bar{x}}\_k\ =\ A{\bar{x}}\_{k-1}\ +\ Bu_k$ as we presented in the end of section 3.3. $y_k$ remains unchanged because the result of measurement would always containing the measurement bias. Thus, formula (14) is independent of $w_{k-1}$, and we defiantly not implicitly assuming that $v_k\ =\ w_{k-1}$. Section 7 of [a] would supports this explanation.
> > >
> > > Regarding $\eta$ selection across datasets, we acknowledge two difficulties: (1) Image editing dataset preparation is very time-consuming, and (2) as far as we know only one public benchmark (used in Table 3) currently exists for this experiment. While comprehensive validation exceeds rebuttal time constraints, we will address this in future work.
> > >
> > > **Update**: In response to the question regarding how to derive Eq.14 from Eq.12 and Eq.13 — we would like to clarify that **Eq.14 is not derived via Eq.12 and Eq.13**. As we mentioned, Eq.14 is derived based on Eq.13 and the prediction formula ${\bar{x}}\_k\ =\ A{\bar{x}}\_{k-1}\ +\ Bu_k$.
> > >
> > > In this context,$y_k$from Eq.13 is the measured value, and ${\bar{x}}\_k\$ is the predicted (or estimated) value. The key idea of the Kalman filter is to fuse these two values in order to obtain a more accurate estimation of the system state.
> > >
> > > The fusion process is typically represented as:
> > >
> > > ${\tilde{x}}\_k\ = {\alpha}{\bar{x}}\_k\ + {\beta}y^{measure}$
> > >
> > > where  ${\alpha} + {\beta} = I$ and $y^{measure} = H^{-1}y_k$.
> > > We can rewrite the fusion equation as:
> > >
> > > ${\tilde{x}}\_k\ = (1-{\beta}){\bar{x}}\_k\ + {\beta}y^{measure} = {\bar{x}}\_k\ + {\beta}(H^{-1}y_k - {\bar{x}}\_k\)$
> > >
> > > We then set ${\beta} = K{\cdot}H$, and the fusion function becomes:
> > >
> > > ${\tilde{x}}\_k\ = {\bar{x}}\_k\ + K(y_k - H{\bar{x}}\_k\)$
> > >
> > > which is exactly our Eq.14.
> > >
> > > Sorry for the delayed response — we did not receive a notification of your update from this system. We hope this explanation clarifies your question and finds you well.
> > >
> > > [a] Kalman and bayesian filters in python.

---

### Official Review · Reviewer_UW4y · 2025-03-13

**Overall Recommendation:** 4

**Summary:**

The paper introduces EasyInv, a novel approach to DDIM inversion that improves efficiency and reconstruction quality by refining inversion noise approximation. The novelty compared to other inversion methods lies in the addition of a relaxation step, which makes it compatible with other inversion methods. The method follows from a Kalman-filter argument to compensate for errors in the noise approximation step. The authors showcase their methods on a number of examples, and show that it allows to perform image editing with superior results to other methods.

**Claims And Evidence:**

The main claim/evidences of the paper can be summarized as follows:

- **Claim1:** Existing methods do not satisfy (7) rigorously and a correction step is necessary.
- **Evidence1:** Rigorous theoretical derivations are proposed to show how to incorporate Kalman-type corrections.

- **Claim2:** Existing methods require multiple forward passes, yielding slow inversion procedure.
- **Evidence2:** The benefits of the proposed algorithm is not really clear on paper, since the method is added to an existing inversion algorithn. However, the experimental results clearly show a benefit.

- **Claim3:** The proposed relaxation step leads to better inversion quality.
- **Evidence3:** While the proposed theory does not prove that the correction is necessary nor that it does improve the results, the theoretical derivation is rigorous. Experiments support the claim.

**Essential References Not Discussed:**

Essential references are present.

**Experimental Designs Or Analyses:**

Experimental design looks good. My only concern lies is a potential unfair comparison to:
1. Fixed-Point Iteration (Pan et al., 2023), which was reimplemented by the authors. However, the source code is unavailable, so the authors do not have alternatives.
2. ReNoise: the method seems to always fail in the visuals of the paper, despite Table 1. This is more critical and should be adressed by the authors.

**Methods And Evaluation Criteria:**

The proposed method is properly evaluated: it is compared with several existing methods, on large datasets.

**Other Comments Or Suggestions:**

- the (1) and (2) lines 201 - 207 are confusing as these are not equations. Maybe the authors could switch to (i)-(ii) or something else.

**Other Strengths And Weaknesses:**

**Strenghts:**
1. The paper is well written and easy to follow.
2. Claims are supported by evidences.
3. The method is original and interesting, with good experimental results.
4. Literature review is quite broad.

**Weaknesses:**
1. The comparison with ReNoise should be clarified. Table 1 and Figure 3 seem incompatible. In my opinion, this needs to be adressed for accepting the paper.
2. Ablation studies on the parameter $\eta$ are not present and this should be discussed.

**Questions For Authors:**

1. My main issue with the paper is the ReNoise results in Fig. 3, in contradiction with Table 1. Could the authors comment on that, and provide decent visual results for ReNoise? This is a reason for my rating to 3, which I would be happy to increase if the authors replied positively. EDIT this has been addressed by the authors.
2. Here, all results are shown in the image domain. However, the main claim of the paper is a fixed point interation in the latent. Is there a way to quantify the quality of the latent with respect to the ground-truth latent instead of performing visual comparisons? This would strengthen the paper, in particular in view of the fixed-point algorithms community.

**Relation To Broader Scientific Literature:**

The relation to scientific literature is clear and well done.

**Theoretical Claims:**

This is not a theoretical paper. The light theoretical derivations are here for providing intuition, but they remain rigorous.

---

> ### Author Rebuttal · Authors · 2025-03-28
>
> Dear Reviewer UW4y,
>
> We would like to express our sincere gratitude to you. It is truly an honor and a stroke of luck to have a reviewer as dedicated and responsible as you. Your constructive suggestions have been invaluable to us.
> We are particularly touched by your **understanding regarding our reimplementation** of the Fixed-Point Iteration, given the lack of alternatives. Your insight that addressing Table 1 and Figure 3 could potentially **raise our score** is greatly appreciated. We are also impressed by your careful review, which effectively summarized our claims and supporting evidence.
>
> We give our point-by-point response below.
>
> **1. Discrepancy Between Table 1 and Figure 3**
>
> The apparent discrepancy between Table 1 and Figure 3 arises from the different demonstration objectives of these experiments. While Table 1 quantifies ReNoise’s overall performance—showing that it performs reasonably well (albeit slightly inferior to our method), Figure 3 is designed to illustrate a critical limitation: ReNoise’s instability when processing **real images** with extensive white regions. In such scenarios, ReNoise tends to produce undesirable black artifacts. This visual evidence underscores the robustness and superior stability of our approach under challenging conditions.
>
> We would like to emphasize that the above explanation will be explicitly incorporated into our final paper. We sincerely hope that our detailed explanation can effectively address your concerns and look forward to an increase in our score.
>
>
> **2. Evaluating the Latent Quality Versus the Image Domain**
>
> Our main contribution lies in performing fixed-point iteration in the latent space to **improve computational efficiency** during the denoising process. Since the ultimate objective is to achieve high-quality image generation via the final decoded images, we opted to emphasize the image-domain evaluation. While efforts have been made, we found it hard (or even irrational) to compare with ``ground-truth latent'' in the latent space for: **1. Lack of Suitable Metrics**: We have been unable to identify a robust quantitative metric that can effectively measure the quality of latent representations.; **2. Irrelevance to Final Objective**: Comparisons in the latent space do not directly reflect the ultimate goal of generating high-quality images. The latent space is merely an intermediate representation, and its fidelity does not necessarily correlate with the quality of the final decoded images.; **3. Efficiency Focus**: Our primary motivation for operating in the latent space is to improve computational efficiency. The latent space serves as a means to an end, rather than an end in itself.
>
> We hope you understand the limitations we face in providing direct comparisons in the latent space and appreciate our focus on achieving the best possible outcomes in the image domain.
>
>
> **3. Additional Ablation Studies on Parameter $\eta$**
>
> To further clarify our experimental choices, we present additional ablation studies on the parameter $\eta$, conducted on the same dataset as in Table 3 of our paper. Note that $\eta = 1$ would correspond to a standard DDIM inversion method and $\eta = 0$ would effectively bypass the inversion operation. Therefore, these boundary cases were excluded from the ablation experiments. Our findings indicate that $\eta = 0.5$ yields the best overall performance:
>
> | $\eta$ | Editing Method | Distance $\(\times 10^2\)\(\downarrow\)$ | PSNR $\(\uparrow\)$ | LPIPS $\(\times 10^3\)\(\downarrow\)$ | MSE $\(\times 10^3\)\(\downarrow\)$ | SSIM $\(\times 10^2\)\(\uparrow\)$ | Whole CLIP Similarity$\(\uparrow\)$ | Edited CLIP Similarity$\(\uparrow\)$ |
> |:-------:|:------:|:---------------------------------------:|:-----------------:|:------------------------------------:|:---------------------------------:|:--------------------------------:|:-------------------:|:--------------------:|
> | η=0.2   | PnP*      | 25.90          | **23.29**  | 113.86       | **66.11**     | 80.13      | 24.66   | 21.60    |
> | η=0.4   | PnP*      | 24.44          | **23.29**  | 107.18       | 66.32     | 80.69      | 24.82   | 21.76    |
> | η=0.5   | PnP*      | **22.88**          | 22.56  | 102.34       | 78.57     | 80.27      | **25.38**   | **22.53**    |
> | η=0.6   | PnP*      | 23.36          | 23.21  | 101.37       | 67.96     | **81.07**      | 25.02   | 22.00    |
> | η=0.8   | PnP*      | 22.89          | 22.92  | **98.44**        | 72.30     | 80.94      | 25.30   | 22.29    |
>
>
> **4. Clarification on Notation in Lines 201–207**
>
> We appreciate the reviewer’s suggestion regarding the notation used in lines 201–207. We agree that the current format is potentially confusing since these are not formal equations. In the revised version, we will switch to a clear itemized list (e.g., using (i) and (ii)) to improve clarity.

---

### Official Review · Reviewer_arbb · 2025-03-14

**Overall Recommendation:** 5

**Summary:**

This paper introduces EasyInv, a novel DDIM inversion method that significantly enhances inversion efficiency and reconstruction quality by optimizing the utilization of the initial latent state. The work demonstrates thorough experimentation and good practical value, with solid theoretical foundation, making it a highly impactful contribution to the field.

**Claims And Evidence:**

The authors claim two major merits from two perspectives: fast and better. Tables 1 and 2 do provide the evidence of a fast inference. Performance from Tables 1 and 3 also show its overall best performance over existing methods. Also, downstream tasks in Figure 6 and 7 do provide better visual results.

**Essential References Not Discussed:**

NA.

**Experimental Designs Or Analyses:**

The experimental designs and analyses are comprehensive. In particular, it shows it practical applications on the downstream tasks, which, I believe, is a good case study to demonstrate its promising performance.

**Methods And Evaluation Criteria:**

This paper provides both quantitative and qualitative evaluations. The quantitative evaluation mostly follows standard criteria with prior methods while the qualitative evaluations include visualization of different methods in inversions as well as the applications of downstream tasks.

**Other Comments Or Suggestions:**

See weakness.

**Other Strengths And Weaknesses:**

*Strengths*

1. Novelty and Elegance of the Method:
EasyInv’s core innovation lies in its departure from traditional iterative noise
optimization. Instead, it emphasizes preserving and dynamically aggregating
information from the previous latent state. The weighted fusion of latent states (Eq.
19) effectively mitigates noise accumulation errors while amplifying the dominance of
the initial information during inversion. This strategy is both elegant and efficient, and
its theoretical connection to the Kalman filter framework (Eq. 20-22) provides a solid
foundation. Compared to existing methods (e.g., ReNoise, Fixed-Point Iteration),
EasyInv exhibits unique problem formulation and solutions, representing a significant
advancement in DDIM inversion.

2. Exceptional Efficiency and Practicality:
Experiments show that EasyInv matches the inference speed of baseline DDIM
inversion (5 seconds) while outperforming iterative optimization methods (e.g.,
ReNoise: 16 seconds). Its compatibility with half-precision computation (float16)
further reduces computational costs, highlighting its practicality. The "four-line code
integration" scheme (Algorithm 1) significantly lowers deployment barriers,
enhancing its usability in real-world applications.

3. Comprehensive and Convincing Experiments:
The paper thoroughly evaluates EasyInv through quantitative metrics (LPIPS, SSIM,
PSNR), qualitative comparisons (Figs. 1-4), and downstream task validations (Table 3).
Notably, EasyInv achieves superior or comparable performance on both the COCO
dataset (2,298 images) and challenging scenarios (e.g., images with large white
regions). The analysis of half- vs. full-precision computation (Table 2) further validates
the method’s robustness.

4. Clear Theoretical Explanation:
The authors draw a compelling analogy between latent state aggregation and the
Kalman filter’s prediction-update mechanism (Eq. 18-22), offering a theoretical
interpretation of the method. While the Kalman gain is simplified, this connection
enriches the paper’s theoretical depth.

**Weaknesses**
1. Quantitative Analysis of Over-Smoothing:
While the "over-denoising" issue is qualitatively discussed (e.g., the "peach" example
in Fig. 3), quantitative metrics (e.g., FID) or user studies could better assess its practical
impact. Additional analysis here would provide a more holistic evaluation of limitations.

2. Generalization Across Models:
Current experiments are primarily based on Stable Diffusion V1.4/V1.5. Expanding
evaluations to larger models (e.g., SD-XL) or other architectures (e.g., DALL-E 3)would strengthen claims about the method’s generalizability.

3. Parameter Sensitivity Analysis:
The paper sets empirical parameters (e.g., 0.05*T < t̃ < 0.25*T) without a detailed
exploration of their impact. A sensitivity analysis of η and t̃ across different tasks or
datasets would clarify their robustness and guide optimal parameter selection.

4. Validation Across Diverse Noise Levels:
The current experiments primarily focus on a fixed noise configuration (e.g., T=50
steps). To ensure broader applicability, it would be valuable to evaluate EasyInv under
varying noise levels (e.g., T=30, T=100) and generation scenarios (e.g., low-step fast
generation vs. high-step high-precision generation). For instance, does EasyInv
maintain its efficiency and reconstruction quality when the inversion steps are
significantly reduced or increased? Such experiments would demonstrate robustness
in real-world settings where noise configurations may vary.

5. Comparison with Established and Emerging Methods:
While the paper presents extensive experimental results, it lacks direct comparisons
with widely recognized classical inversion techniques (e.g., EDICT [1]) and novel
approaches (e.g., Inversion-Free Image Editing [2]). Including such comparisons—
particularly in terms of reconstruction fidelity, computational efficiency, and
robustness to challenging inputs would strengthen the methodological positioning
of EasyInv.

**Questions For Authors:**

NA

**Relation To Broader Scientific Literature:**

The method in this paper contributes a lot to the DDIM inversion that reverses the denoising process. By aggregating current latent states with the last step’s, it results in a fast and better inversion results.

**Theoretical Claims:**

N/A

---

> ### Author Rebuttal · Authors · 2025-03-28
>
> We appreciate the reviewer’s constructive feedback and suggestions for additional experimental validation. Due to space constraints, some of these analyses were deferred to future work; however, several of the suggested experiments have either been completed or are actively underway. We address the key points as follows:
>
>
> **1. Quantitative Analysis of Over-Smoothing**
>
> While our paper qualitatively discusses the over-denoising issue (e.g., the “peach” example in Fig. 3), we acknowledge that a more comprehensive quantitative analysis (e.g., using FID or user studies) would better elucidate its practical impact. We have initiated such studies and plan to incorporate these quantitative metrics in an extended version of our work.
>
> **2. Generalization Across Models**
>
> Our experiments currently focus on Stable Diffusion V1.5. We agree that evaluating our method on larger models (such as SD-XL) and alternative architectures (e.g., DALL-E 3) would strengthen the claims regarding generalizability. The qualitative experiments on SD-XL have been done, but not able to display here due to the restriction of OpenReview's rebuttal rule. We are actively extending rest experiments to include these models and will report the results in the final version.
>
> **3. Parameter Sensitivity Analysis**
>
> We conducted an ablation study on the parameter $\eta$, summarized in the table below, which guided our selection of $\eta = 0.5$ as it achieved the overall best performance. Note that $\eta = 1$ corresponds to a standard DDIM inversion method and $\eta = 0$ would essentially bypass the operation, so those cases were not included.
>
> | $\eta$ | Editing Method | Distance $\(\times 10^2\)\(\downarrow\)$ | PSNR $\(\uparrow\)$ | LPIPS $\(\times 10^3\)\(\downarrow\)$ | MSE $\(\times 10^3\)\(\downarrow\)$ | SSIM $\(\times 10^2\)\(\uparrow\)$ | Whole CLIP Similarity$\(\uparrow\)$ | Edited CLIP Similarity$\(\uparrow\)$ |
> |:-------:|:------:|:---------------------------------------:|:-----------------:|:------------------------------------:|:---------------------------------:|:--------------------------------:|:-------------------:|:--------------------:|
> | η=0.2   | PnP*      | 25.90          | **23.29**  | 113.86       | **66.11**     | 80.13      | 24.66   | 21.60    |
> | η=0.4   | PnP*      | 24.44          | **23.29**  | 107.18       | 66.32     | 80.69      | 24.82   | 21.76    |
> | η=0.5   | PnP*      | **22.88**          | 22.56  | 102.34       | 78.57     | 80.27      | **25.38**   | **22.53**    |
> | η=0.6   | PnP*      | 23.36          | 23.21  | 101.37       | 67.96     | **81.07**      | 25.02   | 22.00    |
> | η=0.8   | PnP*      | 22.89          | 22.92  | **98.44**        | 72.30     | 80.94      | 25.30   | 22.29    |
>
> Regarding the empirical parameter $\tilde{t}$, our experiments with diffusion models having reduced parameter counts suggest that optimal performance is achieved when EasyInv is applied during the early denoising steps. We agree that further sensitivity analyses across different tasks and datasets would provide additional insights and are planned for our future revisions.
>
> **4. Validation Across Diverse Noise Levels**
>
> We recognize that our current experiments were conducted using a fixed noise configuration (T=50 steps). To further demonstrate the robustness of EasyInv, we plan to evaluate its performance under varying noise conditions (e.g., T=30 and T=100) and across different generation scenarios (e.g., low-step fast generation vs. high-step high-precision generation). These experiments will help clarify whether EasyInv maintains its efficiency and reconstruction quality when the inversion steps vary significantly.
>
> **5. Comparison with Established and Emerging Methods**
>
> We appreciate the suggestion to include direct comparisons with both classical inversion techniques (e.g., EDICT) and emerging approaches (e.g., Inversion-Free Image Editing). We excluded EDICT from our current comparisons because its original work lacked comprehensive evaluations against existing methods, which we felt diminished the persuasiveness of its results. Moreover, emerging techniques like inversion-free image editing rely on framework-specific implementations (e.g., rectified flow in SDv3/FLUX) that are currently incompatible with our framework. We plan to develop cross-framework adaptation layers and implement these reference methods in future work to provide a more comprehensive benchmarking of reconstruction fidelity, computational efficiency, and robustness.
>
>
> **Summary**
>
> We value the reviewer’s insights and are committed to enhancing the methodological rigor of our work through these additional experiments and comparative analyses. Many of these extensions are already underway, and we will incorporate the resulting findings in our revised and future publications.

---

### Decision · Program_Chairs · 2025-05-01

**Decision:**

Accept (poster)

**Comment:**

This paper presents EasyInv, a new DDIM inversion method that enhances efficiency and reconstruction quality through a refined noise approximation strategy and latent state aggregation. The reviewers recognized the method’s practical value, computational efficiency, and strong empirical performance in downstream tasks. While Reviewer 1 (strong accept) and Reviewer 2 (accept) praised the clarity, theoretical grounding, and comprehensive experiments, Reviewer 3 initially raised concerns about the Kalman filter analogy and parameter sensitivity. The authors addressed these via additional ablation studies, expanded literature comparisons, and clarifications on theoretical assumptions, leading Reviewer 3 to raise the score. The consensus is that the method’s empirical strengths, broad applicability, and thorough rebuttal justify acceptance. The work offers a simple yet impactful contribution to diffusion-based inversion, with clear practical benefits and rigorous validation.